# A Novel LRRK2 Variant p.G2294R in the WD40 Domain Identified in Familial Parkinson’s Disease Affects LRRK2 Protein Levels

**DOI:** 10.3390/ijms22073708

**Published:** 2021-04-02

**Authors:** Jun Ogata, Kentaro Hirao, Kenya Nishioka, Arisa Hayashida, Yuanzhe Li, Hiroyo Yoshino, Soichiro Shimizu, Nobutaka Hattori, Yuzuru Imai

**Affiliations:** 1Department of Research for Parkinson’s Disease, Juntendo University Graduate School of Medicine, Tokyo 113-8421, Japan; j-ogata@juntendo.ac.jp (J.O.); nhattori@juntendo.ac.jp (N.H.); 2Department of Geriatric Medicine, Tokyo Medical University, 6-7-1 Nishishinjuku, Shinjuku-ku, Tokyo 160-0023, Japan; kentaroh@tokyo-med.ac.jp (K.H.); soichiro@tokyo-med.ac.jp (S.S.); 3Department of Neurology, Juntendo University School of Medicine, 2-1-1 Hongo, Bunkyo-ku, Tokyo 113-8421, Japan; nishioka@juntendo.ac.jp (K.N.); arisa-h@juntendo.ac.jp (A.H.); yuanzhe@juntendo.ac.jp (Y.L.); 4Research Institute for Diseases of Old Age, Graduate School of Medicine, Juntendo University, 2-1-1 Hongo, Bunkyo-ku, Tokyo 113-8421, Japan; yhiroyo@juntendo.ac.jp

**Keywords:** familial Parkinson’s disease, *LRRK2*, genetics, macrophage, loss-of-function

## Abstract

*Leucine-rich repeat kinase 2* (*LRRK2*) is a major causative gene of late-onset familial Parkinson’s disease (PD). The suppression of kinase activity is believed to confer neuroprotection, as most pathogenic variants of *LRRK2* associated with PD exhibit increased kinase activity. We herein report a novel *LRRK2* variant—p.G2294R—located in the WD40 domain, detected through targeted gene-panel screening in a patient with familial PD. The proband showed late-onset Parkinsonism with dysautonomia and a good response to levodopa, without cognitive decline or psychosis. Cultured cell experiments revealed that p.G2294R is highly destabilized at the protein level. The LRRK2 p.G2294R protein expression was upregulated in the patient’s peripheral blood lymphocytes. However, macrophages differentiated from the same peripheral blood showed decreased LRRK2 protein levels. Moreover, our experiment indicated reduced phagocytic activity in the pathogenic yeasts and α-synuclein fibrils. This PD case presents an example wherein the decrease in LRRK2 activity did not act in a neuroprotective manner. Further investigations are needed in order to elucidate the relationship between LRRK2 expression in the central nervous system and the pathogenesis caused by altered LRRK2 activity.

## 1. Introduction

Mutations in the genes *leucine-rich repeat kinase 2* (*LRRK2*) and *synuclein alpha* (*SNCA*) were predominantly identified in familial Parkinson’s disease (PD) beyond the countries. *LRRK2* was originally mapped to 12p11.2-13.1 in a large family from Sagamihara in Japan [1]. Four studies identified *LRRK2* variants—p.R1441C, p.Y1699C, p.G2019S, and p.I2020T—in several races [2,3,4,5]. Brain pathology commonly indicated non-specific neuronal loss and gliosis in the substantia nigra, as well as Lewy pathology [6]. Furthermore, some patients showed tau pathology, TDP-43 inclusions, or glial cytoplasmic inclusions [6,7]. Patients with *LRRK2* variants showed middle- or late-onset Parkinsonism, a good response to levodopa, and few complications associated with cognitive decline [8]. The estimated prevalence in our cohort was 1–2%. 

LRRK2 is a large multi-domain protein with 2527 amino acids belonging to the ROCO protein family and consisting of multiple domains, including a leucine-rich repeat, Ras of complex proteins (ROC), C-terminal of Roc (COR), kinase, and WD40 repeat domains. Previous studies have reported that pathogenic mutations of LRRK2 cause increased kinase activity, suggesting that abnormalities in the substrate regulation by LRRK2 are involved in the pathogenesis. However, the absence of symptomatic differences between heterozygous and homozygous p.R1441H implies the existence of a complex pathological mechanism [9].

Here, we report a novel LRRK2 variant—p.G2294R—which may affect the structure of the WD40 repeat domain. Our cell culture study found that the p.G2294R variant markedly reduced LRRK2 protein levels, exhibiting a loss-of-function tendency.

## 2. Results

### 2.1. PD Case with LRRK2 p.G2294R

A novel heterologous *LRRK2* variant expected to replace p.G2294 with R in the WD40 repeat domain was found in a patient with PD (Figure 1A). The family was of Japanese origin. The proband with the *LRRK2* variant (II-4) was 70 years old at the time of our examination (Figure 1B). At 67 years of age, she noticed resting tremors in her lower left limb. The patient visited a local hospital, where an attending physician diagnosed her with PD upon finding akinesia, resting tremors, and a good response to levodopa carbidopa. Her symptoms met the standard clinical criteria for PD, categorized as clinically-established PD [10]. She did not present with cognitive decline, rigidity, or gait disturbance at the first neurological examination. Regarding her family history, her cousin (II-8) developed PD and died at 70 years of age (Figure 1B). The clinical details of II-8 are unknown. The proband (II-4) had a monozygotic twin sister (II-6) who did not show any Parkinsonism or other symptoms related to neurological disorders at 70 years of age (Figure 1B). The peripheral blood from the patient (II-4) indicated a normal white blood cell count, C-reactive protein level, and erythrocyte sedimentation rate. Brain magnetic resonance imaging did not reveal any abnormalities. [^123^I]N-ω-fluoropropyl-2b-carbomethoxy-3β-(4-iodophenyl) tropane [^123^I]FP-CIT single photon emission computed tomography indicated a decrease in dopamine transporter (specific binding ratio: right, 3.07; left, 3.14). [^123^I]meta-iodobenzylguanidine myocardial scintigraphy showed a decrease in the heart to mediastinum ratio (early 1.37, a delay of 1.19 in the heart to mediastinum ratio). The following year, she noticed a loud voice while sleeping. At 70 years of age, she visited our hospital and complained of gait disturbance and pain in her toes and lower limbs, off-period every evening. She was prescribed 450 mg levodopa carbidopa, 300 mg catechol-O-methyltransferase inhibitor, 4 mg ropinirole, and 10 mg selegiline hydrochloride per day. A neurological examination showed olfactory disturbance, resting tremors, akinesia, retropulsion, on-off phenomenon, constipation, urinary incontinence, and orthostatic hypotension. Her Movement Disorder Society-Unified Parkinson’s Disease Rating Scale (MDS-UPDRS) score was 13 (Part I, 6; Part II, 1; Part III, 2; Part IV, 4) at the on-period. Her Hoehn and Yahr stage indicated 0 at the on-period and 1 at the off-period. The indices of the scores related to cognitive function were 30 of 30 on the Mini-Mental State Examination, 30 of 30 on the revised Hasegawa’s dementia scale, and 13 of 18 on the Frontal Assessment Battery. She was able to live without any support, with a good response to anti-Parkinsonian drugs.

### 2.2. p.G2294R Causes Reduced LRRK2 Protein Expression

Based on the reported crystal structure of the WD40 repeat domain, p.G2294 may not be located at the interface of dimer formation, but may affect the backbone of the β-propeller structure [11] (Appendix A). This amino acid is located between the thirteenth and fourteenth β-sheet structures, and is highly conserved, at least from human to fish (Appendix A). We examined the effects of p.G2294R replacement on LRRK2 in cultured human cells. We found that the protein levels of FLAG-tagged LRRK2 p.G2294R were lower than those of FLAG-tagged LRRK2 WT (Figure 2A). Using independent expression vectors to produce Myc-tagged LRRK2 and stable cell lines harboring a copy of the FLAG-LRRK2 expression cassette, we confirmed that the p.G2294R variant reduced LRRK2 expression (Figure 2A and Appendix A). The protein expression of LRRK2 p.G2294R—but not WT—was decreased by treatment with the protein synthesis inhibitor cycloheximide, suggesting that LRRK2 p.G2294R decreases protein stability (Figure 2B). The lack of LRRK2 p.G2294R protein-level restoration—either by autophagy or proteasome inhibition—suggested that LRRK2 p.G2294R is subjected to an alternative degradation pathway (Appendix A). As a result of LRRK2 p.G2294R reduction, the phosphorylation of Rab10—a known LRRK2 substrate—was also reduced (Figure 2A and Appendix A). A similar result was obtained in the phosphorylation of Rab8, another LRRK2 substrate (Figure 2A).

LRRK2 regulates the phagocytosis of macrophages [12,13]. We collected PBMCs from the proband (II-4) and a healthy control, and further isolated CD14^+^ monocytes from PBMCs. The gender-matched control case was 74 years old, and was without neurodegenerative or neurological disorders. In contrast to the results in cultured cells, the LRRK2 protein levels were increased in the PD case with LRRK2 p.G2294R. However, the LRRK2 protein levels were reduced in macrophages differentiated from monocytes with LRRK2 p.G2294R (Figure 2C,D). Accordingly, there was a decreasing trend in the absolute amounts of phosphorylated Rab8 and Rab10 in the macrophages (Figure 2C).

The phagocytic activity of the macrophages was analyzed using zymosan prepared from yeast cell wall as a model of fungi (Figure 3 and Appendix A). The acidification of the phagosomes, as assessed by the uptake of pH-sensitive fluorescent (pHrodo) zymosan bioparticles, was impaired in the macrophages from the PD patient with LRRK2 p.G2294R (Figure 3D). Although the subcellular distribution of LRRK2, Rab8, and LAMP1 was not markedly different between the normal control and LRRK2 p.G2294R (Figure 3A,B), the strong accumulation of phospho-Rab10 signals was frequently observed in the control macrophages but not in the LRRK2 p.G2294R macrophages (Figure 3C).

In order to analyze the phagocytic activity of the macrophages in another pathogenic setting, we employed pH-sensitive fluorescent α-synuclein fibrils as a PD-associated substance (Figure 4A). The phagocytic activity was significantly impaired in the LRRK2 p.G2294R macrophages, suggesting that the phagocytic activity was affected by LRRK2 p.G2294R (Figure 4A,C and Appendix A). However, the lysosomal cathepsin B activity estimated by Magic Red™ substrate was not different between the two, suggesting that the lysosomal activity of macrophages is not affected by LRRK2 p.G2294R (Figure 4B,C).

## 3. Discussion

The proband (II-4) was diagnosed with a common type of PD. Her clinical symptoms were confirmed in other patients with *LRRK2* pathogenic variants such as tremors, akinesia, rigidity, gait disturbance with good response to levodopa, and dysautonomia, without cognitive decline or psychosis [8,9]. In the absence of brain pathology, segregation study data, or the details of another PD patient (II-8) due to her death, the extent to which LRRK2 p.G2294R was involved in the pathogenesis remains unclear. The small number of PD patients in the family and the non-development of Parkinsonism in another monozygotic twin at 70 years (II-6) suggest a low penetrance of p.G2294R. The monozygotic twin (II-6) may develop Parkinsonism in the future, as patients with *LRRK2* variants sometimes develop Parkinsonism later in life. The estimated age at onset of the patients with *LRRK2* variants was 61.4 ± 11.8, with a range of 40–73 years in the Japanese cohort [8]. Reportedly, the penetrance ratio of LRRK2 p.G2019S is 42.5% in non-Ashkenazi Jews and 26% in Ashkenazi Jews, among PD patients [15,16]. Thus, it may be reasonable to conclude that p.G2294R shows low penetrance. 

The LRRK2 WD40 repeat domain is required for kinase activity [17], and is associated with synaptic vesicles [18] and microtubules [19], contributing to its neurotoxicity [17,20]. In contrast, variant G2385R in the WD40 repeat domain, which is associated with an increased risk of PD in Taiwanese Chinese and other Asian populations [21], confers partial loss-of-function in terms of synaptic vesicle trafficking [22]. In this study, LRRK2 p.G2294R revealed interesting protein properties. p.G2294R drastically reduced the expression of the LRRK2 protein. Although the major proteolytic pathways—the proteasome and autophagy pathways—are not actively involved in the reduction of LRRK2 expression by p.G2294R, the WD40 structure distorted by p.G2294R may destabilize the LRRK2 protein.

The differentiated macrophages from this patient showed reduced phagocytic capacity to pathogens and α-synuclein fibrils. Decreased lysosomal activity has been suggested in *LRRK1^−/−^; LRRK2^−/−^* mice [23], while LRRK2 regulates vesicle sorting from lysosomes [24]. Decreased lysosomal function is presumed if p.G2294R yields a loss of function. However, there was no difference in the cathepsin B activity between the macrophages from the patient and the control, suggesting that the lysosomal functions were intact in the macrophages harboring p.G2294R.

This genetic and clinical study of p.G2294R has limitations due to the lack of pathology and segregation-study data. In our experiment, the PBMCs from patient II-4 showed elevated expression levels of LRRK2. Although LRRK2 is upregulated by INFγ [13,25], no inflammatory profile was found in the PBMCs of this case. Thus, the cause of the elevated LRRK2 expression in PBMCs remains unresolved. If the loss-of-function property of LRRK2—namely, reduced protein levels—is found in the central nervous system with this variant, as observed in our cultured cell study, the suppression of LRRK2 function is unlikely to protect neurons. A recent meta-analysis using a large population and public gene database revealed that loss-of-function variants of LRRK2 are not strongly associated with any specific phenotype of PD patients [26]. Another study also reported no association of LRRK2 loss-of-function between 11,095 PD and 12,615 controls [27]. These large population studies may not support the idea that reduced LRRK2 activity increases the risk of PD. Further studies on p.G2294R—a rare variant which may have implications for future drug discovery strategies against LRRK2—are necessary.

## 4. Materials and Methods

### 4.1. Genetic Screening

DNA was extracted from peripheral blood using the standard method. Targeted gene panels were used for the genetic screening. The details of these methods have been described previously [28]. All of the putative pathogenic variants were confirmed by Sanger sequencing. 

### 4.2. Human Donor Characterization

The PD participants (70-year-old females) and healthy controls (74-year-old females without a familial history of PD) were of Japanese ethnic origin. Whole blood was collected from each individual, and peripheral blood mononuclear cells (PBMCs) from peripheral blood were fractionated using a Vacutainer^®^ CPT™ cell preparation tube (BD Biosciences, Franklin Lakes, NJ, USA; REF 362761) with sodium citrate.

### 4.3. Monocyte Isolation and Differentiation

Monocytes were isolated from the PBMCs using a magnetic cell separation system with anti-CD14 mAb-coated microbeads (Miltenyi Biotec, Bergisch Gladbach, Germany; 130-050-201). The CD14-positive monocytes were differentiated in RPMI 1640 medium (Nacalai Tesque, Inc., Kyoto, Japan; 3026485) supplemented with 10% heat-inactivated fetal calf serum, 50 µM 2-mercaptoethanol, and 50 ng/mL granulocyte-macrophage colony-stimulating factor (Proteintech, Rosemont, IL, USA; HZ-1002). The cells were cultured at 37 °C under a humidified 5% CO_2_ atmosphere for 6 days, and the medium was replaced on day 3. On day 6, the cells were washed and detached with 0.5 mM EDTA in ice-cold PBS and plated in a 96-well glass-bottom microplate at a concentration of 2 × 10^5^ cells/well in RPMI 1640 medium containing 10% heat-inactivated fetal calf serum.

### 4.4. Phagocytosis and Magic Red™ Assays

Recombinant α-synuclein, prepared as described previously [29], was labeled with the pH-sensitive fluorescent dye AcidiFluor™ ORANGE (Goryo Chemical, Inc., Tokyo, Japan; GC303). Non-labeled recombinant α-synuclein (1 mg) was added to AcidiFluor™ ORANGE-labeled α-synuclein (100 µg) in 30 mM Tris-HCl (pH 7.4) buffer, and was incubated at 37 °C with shaking at 250 rpm for 120 h in order to obtain α-synuclein fibrils. The fibrils were sonicated on ice using a high-power sonicator (Taitec Corporation, Saitama, Japan; VP-050N) at a pulse-width modulation of 15–20% for 7 min in order to obtain α-synuclein fibrils of approximately 50 nm in length. The differentiated macrophages were treated with AcidiFluor™ ORANGE-labeled α-synuclein fibrils or pHrodo™ Green Zymosan Bioparticles™ Conjugate for Phagocytosis (Thermo Fisher Scientific, Waltham, MA, USA; P35365) for 2 h. The differentiated macrophages were treated with a Magic Red Cathepsin B Assay Kit (ImmunoChemistry Technologies, Bloomington, MN, USA; 938) for 15 min.

### 4.5. Plasmids and Stable Cell Lines

The LRRK2 p.G2294R mutation was introduced into pcDNA5/FRT-FLAG-LRRK2 [30] and pcDNA5/FRT-Myc-LRRK2 [30] by site-directed mutagenesis. In order to generate stable cell lines expressing one copy of LRRK2 wild-type (WT) or p.G2294R, pcDNA5/FRT vector and Flp recombinase expression vector pOG44 were co-transfected with Flp-In™ 293 T-REx cells using Lipofectamine™ 2000 (Thermo Fisher Scientific, Waltham, MA, USA; K652020, R78007, 11668019). After 2 days of transfection, cells harboring the expression cassette were selected with 100-µg/mL hygromycin B. Single colonies were isolated, and the expression of FLAG-LRRK2 was validated by Western blotting.

### 4.6. Protein Stability Assay

pcDNA5/FRT-FLAG-LRRK2 WT and p.G2294R were transfected into HEK293T cells. After 48 h of transfection, the cells were cultured for 8 h with mock, 100 μL/mL cycloheximide (Sigma-Aldrich, St. Louis, MO, USA; C1988), 10 μM MG-132 (Peptide Institute, Inc., Osaka, Japan; 3175-v), or 100 nM bafilomycin A1 (Sigma-Aldrich; SML1661). The LRRK2 protein levels were detected by Western blotting.

### 4.7. Antibodies for Immunofluorescence and Western Blotting

The following primary antibodies were used: rabbit anti-LRRK2 (Abcam, Cambridge, UK; ab133474; 1:100 dilution for immunofluorescence (IF) and 1:1000 for western blotting (WB)), mouse anti-Rab8 (BD; 610844; 1:200 for IF and 1:2000 for WB), rabbit anti-Rab8 phospho-T72 (Abcam; ab230260; 1:500 for WB), mouse anti-LAMP1 (DSBH, Iowa City, IA, USA; H4A3; 1:50 for IF), rabbit anti-Rab10 (Abcam; ab237703; 1:100 for IF and 1:1000 for WB), rabbit anti-Rab10 phospho-T73 (Abcam; ab230261; 1:100 for IF and 1:1000 for WB), anti-FLAG tag (Sigma-Aldrich; F1804; 1:1000 for WB), anti-Myc tag (Cell Signaling Technology, Danvers, MA, USA; 2276; 1:2000 for WB), anti-β-actin (Sigma-Aldrich; A5441; 1:5000 for WB), rabbit anti-p62 (Medical & Biological Laboratories Co., Ltd., Nagoya, Japan; PM045, 1:1000 for WB), and rabbit anti-LC3B (Cell Signaling; 3868, 1:2000 for WB).

### 4.8. Western Blotting

The cells were lysed with cell lysis buffer containing 50 mM Tris pH 7.4, 120 mM NaCl, 5 mM EDTA, 1% Triton-X100, 10% glycerol with a protease inhibitor cocktail, and a phosphatase inhibitor cocktail, and were centrifuged at 15,000× *g* for 10 min at 4 °C. The resultant supernatant was denatured with 3× Laemmli SDS sample buffer containing 15% 2-mercaptoethanol. The samples were separated by SDS-PAGE and transferred onto polyvinylidene fluoride membranes. The membranes were blocked for 1 h with 5% skim milk or 5% bovine serum albumin (Rab10 phospho-T73 antibody), and then incubated with primary antibodies for 1 h at 22 °C or overnight at 4 °C. After washing 3 times with Tris-buffered saline (TBS) at pH 7.4 containing 0.05% Tween 20, the membranes were incubated with HRP-conjugated secondary antibodies for 1 h at 22 °C, and were again washed three times with TBS-Tween 20. The blots incubated with Immobilon Forte Western HRP substrate (Merck Millipore, Burlington, MA, USA; WBLUF0100) were detected using a FUSION FX chemiluminescence imaging system (VILBER, Paris, France). The blot intensities were quantitated using Image Lab Software Version 6.0.0 (Bio-Rad Laboratories, Hercules, CA, USA).

### 4.9. Statistical Analysis

A two-tailed Student’s *t*-test was used to determine the significant differences between the two groups. The data distribution was assumed to be normal, but this was not formally tested. Randomization was used for each genotype, and the data collection and analysis were not performed blind to the experimental conditions.

## Figures and Tables

**Figure 1 ijms-22-03708-f001:**
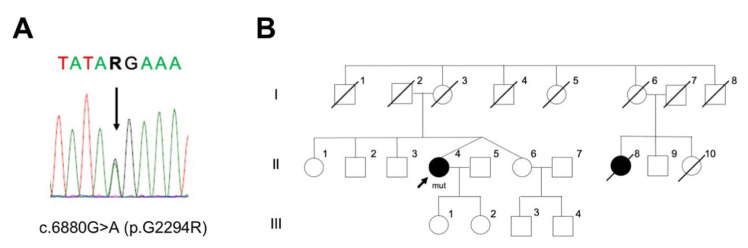
PD case with *LRRK2* p.G2294R and her family pedigree. (**A**) Sanger sequencing found a heterozygous missense variant, NM_198578.4: c.6880G > A (p.G2294R). A, Adenine; T, Thymine; G, Guanine; R, purine (A or G). (**B**) The family pedigree of a patient with LRRK2 p.G2294R. Black arrow, proband; square, male; circle, female; oblique line, deceased; black-colored circles, clinically diagnosed with PD. II-4 and II-6 were monozygotic twins. II-8 was not assessed by genetic testing.

**Figure 2 ijms-22-03708-f002:**
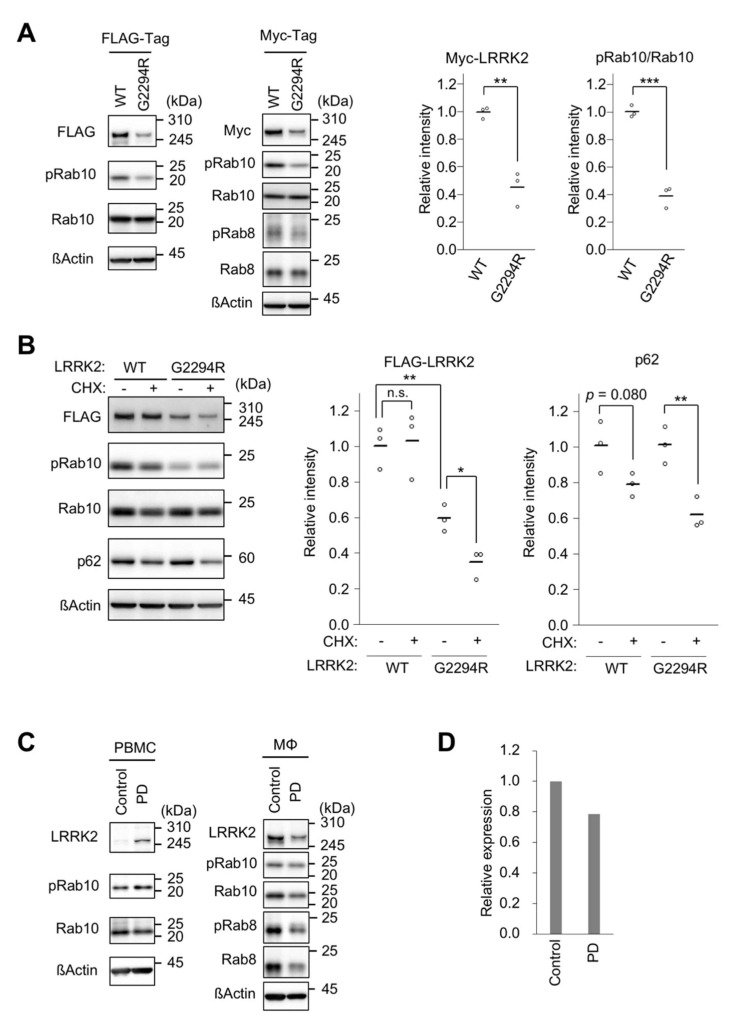
LRRK2 is destabilized by the p.G2294R variant. (**A**) The protein levels of LRRK2. Lysate from HEK293T cells transiently transfected with FLAG-LRRK2 (left) or Myc-LRRK2 (right) was analyzed by Western blot with indicated antibodies. The graphs (bars represent the mean; white circles indicate each value) represent relative Myc-LRRK2 levels normalized with ß-actin, and the ratio of pRab10 to total Rab10. Note that the anti-pRab8 we used crossreacts with pRab3, pRab10, and pRab35 [14]. *n* = 3 biological replicates; Student’s *t*-test. ß-actin served as a loading control. pRab10 and Rab10 phosphorylated at T73; pRab8, Rab8 phosphorylated at T72. (**B**) LRRK2 p.G2294R is unstable. HEK293T cells transiently transfected with FLAG-LRRK2 WT or p.G2294R were treated with (+) or without (−) 100 µg/mL cycloheximide (CHX) for 8 h and analyzed by Western blot. The left graph shows that the relative LRRK2 intensity normalized with ß-actin was decreased by p.G2294R mutation, which was further accelerated by CHX treatment. The right graph shows the relative p62 intensity normalized with ß-actin. The data (bars represent the mean; white circles indicate each value) are from 3 biological replicates; Student’s *t*-test. n.s., not significant. (**C**) The endogenous protein levels of LRRK2 and the indicated proteins in PBMCs and macrophages (Mø) from the PD patient with LRRK2 p.G2294R, and a normal control. (**D**) The relative quantification of endogenous LRRK2 normalized with ß-actin in Mø.

**Figure 3 ijms-22-03708-f003:**
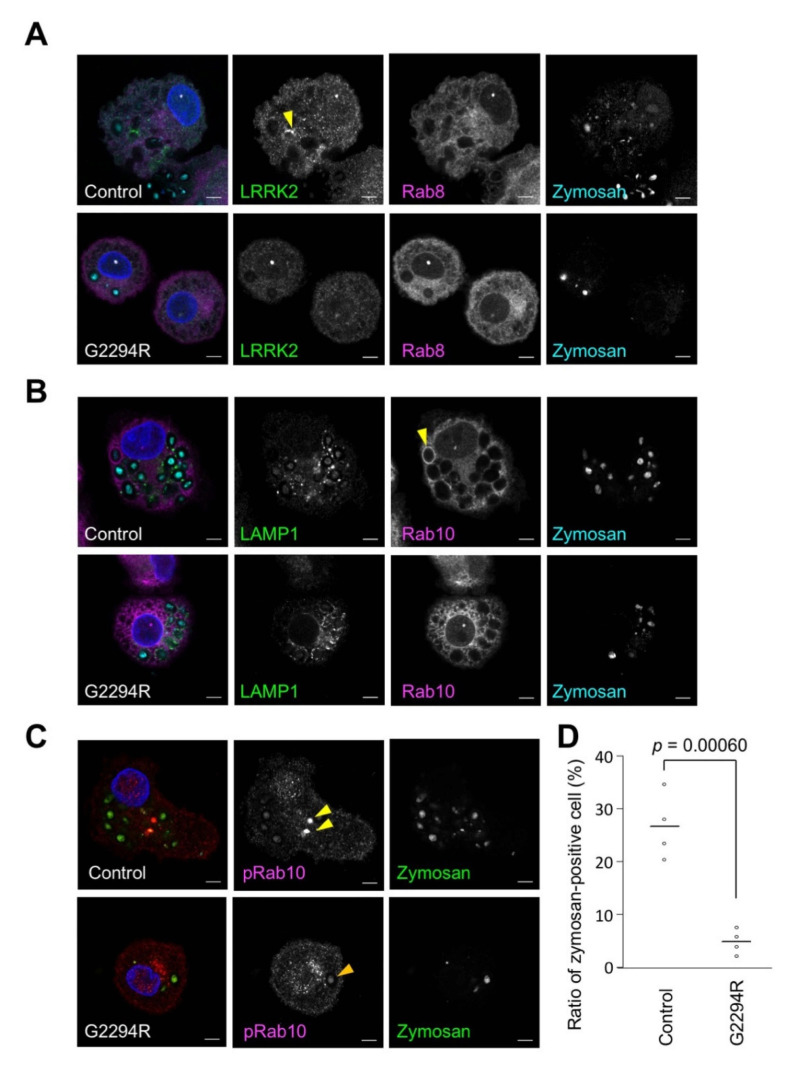
Phagocytosis is impaired by the p.G2294R variant. (**A**–**C**) Macrophages differentiated from monocytes were treated with zymosan conjugated with pHrodo Green for 2 h. The subcellular localization of the indicated endogenous proteins and nuclei (blue) was visualized with specific antibodies and DAPI, respectively. The scale bars are 5 µm. The yellow arrowheads in (**A**,**B**) indicate LRRK2 and Rab10 accumulated on phagosome membranes, respectively. (**C**) The yellow arrowheads in the control cells indicate phosphorylated Rab10 strongly accumulated at the zymosan, and the orange arrowhead in p.G2294R cells shows phosphorylated Rab10 weakly accumulated at the zymosan. (**D**) The graph shows the ratio of zymosan-positive cells, which was decreased by p.G2294R mutation. The data (bars represent the mean, *n* ≥ 52) are from four biological replicates; Student’s *t*-test.

**Figure 4 ijms-22-03708-f004:**
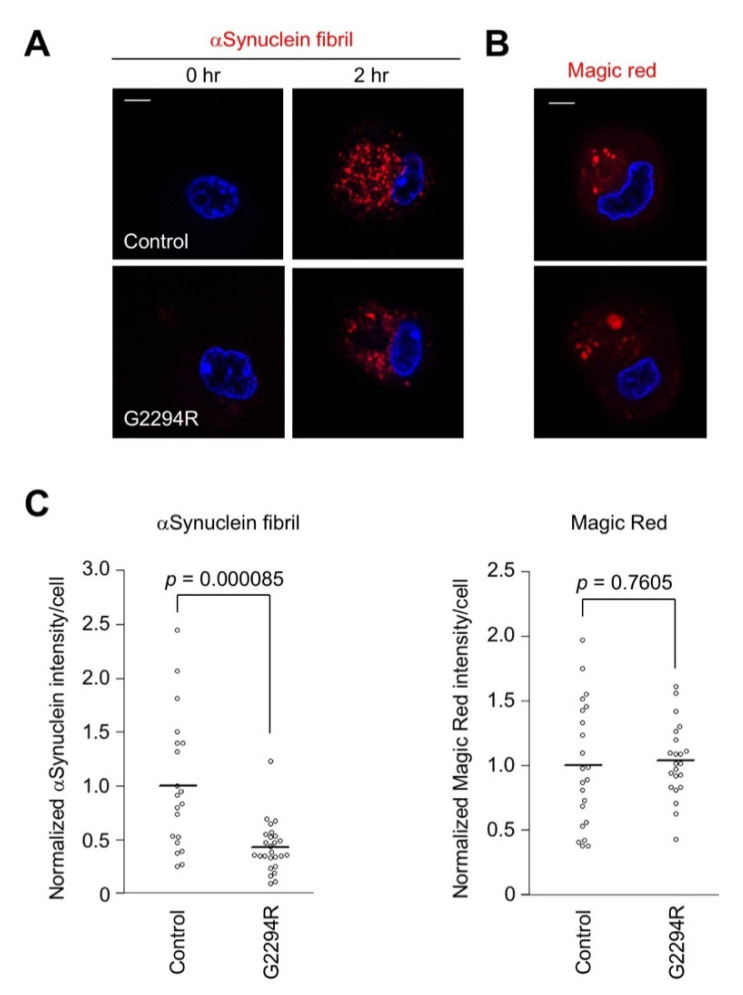
Uptake of α-synuclein fibrils is impaired by p.G2294R variant. (**A**) Macrophages differentiated from monocytes were treated with α-synuclein fibrils conjugated with AcidiFluor ORANGE for 2 h. The fluorescence of α-synuclein fibrils activated in low-pH endolysosomal vesicles (red) and nuclei (blue) was imaged. The scale bar is 5 µm. (**B**) The cathepsin B activity (red) in the macrophages was imaged using Magic Red substrate. The scale bar is 5 µm. (**C**) The quantification of phagocytosed α-synuclein fibrils and cathepsin B activity. The bars in the graphs represent the mean (*n* ≥ 20); Student’s *t*-test.

## Data Availability

The data presented in this study are available on request from the corresponding author.

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
