# Peer review of "A Novel LRRK2 Variant p.G2294R in the WD40 Domain Identified in Familial Parkinson’s Disease Affects LRRK2 Protein Levels"

_ijms, 2021, doi:10.3390/ijms22073708_

Round 1

Reviewer 1 Report

Leucine-rich repeat kinase 2 (LRRK2) is a major causative gene of late-onset familial Parkinson's disease (PD). Most pathogenic variants of LRRK2 associated with PD exhibit increased kinase activity. A novel LRRK2 variant—p.G2294R—located in the WD40 domain identified in a patient with familial PD. p.G2294R causes reduced LRRK2 protein expression. Moreover Phagocytosis is impaired by p.G2294R variant.  The study is comprehensive and well designed.  The data presented well.

  1. LRRK2 p.G2294R variant leads reduction of phosphor Rab10. Have the authors verified other Rab family members.
  2. α-synuclein fibrils uptake is impaired by p.G2294R variant. It’s hard to believe that, authors may include the larger field image in the supplementary figure.

Author Response

1. LRRK2 p.G2294R variant leads reduction of phosphor Rab10. Have the authors verified other Rab family members.

We checked phospho-Rab8 (T72) levels in HeLa cells transiently expressing LRRK2 and macrophages from the patient although the commercially-available antibody we used cross-react with phospho-Rab3, phospho-Rab10 and phospho-Rab35 (Lis et al., 2018). The fact was mentioned in the figure legend.

Reference: Lis, P., Burel, S., Steger, M., Mann, M., Brown, F., Diez, F., Tonelli, F., Holton, J.L., Ho, P.W., Ho, S.L., et al. (2018). Development of phospho-specific Rab protein antibodies to monitor in vivo activity of the LRRK2 Parkinson's disease kinase. Biochem J 475, 1-22.

2. α-synuclein fibrils uptake is impaired by p.G2294R variant. It’s hard to believe that, authors may include the larger field image in the supplementary figure.

Because it was difficult to detect the weak fluorescence signals of α-synuclein fibrils at a lower magnification, we could not take their larger field images. Instead, all raw images used for quantification are inserted as supplemental data (Supplemental Figure S4).

Reviewer 2 Report

This is a valuable study on LRRK2 protein levels and its loss-of-function tendency in a novel LRRK2 variant p.G2294R in the WD40 domain identified in familial Parkinson's disease.

Since LRRK2 is linked to innate immunity and the immune system and inflammation potentially important for PD progression, it is recommended to address the two below issues:

  1. What data support this statement: "no inflammatory profile was found in the PBMCs of this case "?
  2. It is also recommended to provide results of the analysis of protein levels of LRRK2, Rab10 and Rab10 phosphorylated at T73 in neutrophils to Figure 2 C and D.

Author Response

1. What data support this statement: "no inflammatory profile was found in the PBMCs of this case "?

We described supported results in the Result section as follows:

(Line 67) Peripheral blood from the patient (II-4) indicated normal white blood cell count, C-reactive protein level, and erythrocyte sedimentation rate. Brain magnetic resonance imaging did not reveal any abnormalities.

2. It is also recommended to provide results of the analysis of protein levels of LRRK2, Rab10 and Rab10 phosphorylated at T73 in neutrophils to Figure 2C and D.

Thank you for pointing out the fact that LRRK2 is highly expressed on neutrophils as well as monocytes. In order to make the best use of the patient's limited blood, we used one for monocyte isolation and the other for establishment of iPS cells from lymphocytes. We chose monocytes because prior literature had shown that LRRK2 is involved in phagocytosis of macrophages, and because we thought they would be ideal for functional analysis. On the other hands, since neutrophils have higher protease activity (Fan et al., 2018), we thought that biochemical analysis of isolated neutrophils from the limited resource would be risky. The establishment of iPS cell would allow us to use neurons for functional analysis of pathologies related to PD.

Reference: Fan, Y., Howden, A.J.M., Sarhan, A.R., Lis, P., Ito, G., Martinez, T.N., Brockmann, K., Gasser, T., Alessi, D.R., and Sammler, E.M. (2018). Interrogating Parkinson's disease LRRK2 kinase pathway activity by assessing Rab10 phosphorylation in human neutrophils. Biochem J 475, 23-44.

Round 2

Reviewer 2 Report

N/A